# Proteins and DNA Sequences Interacting with Tanshinones and Tanshinone Derivatives

**DOI:** 10.3390/ijms26020848

**Published:** 2025-01-20

**Authors:** Piotr Szymczyk, Małgorzata Majewska, Jadwiga Nowak

**Affiliations:** 1Department of Biology and Pharmaceutical Botany, Medical University of Lodz, Muszyńskiego 1, 90-151 Lodz, Poland; 2Department of Oncobiology and Epigenetics, Faculty of Biology and Environmental Protection, University of Lodz, Pomorska 141/143, 90-236 Lodz, Poland; malgorzata.majewska@biol.uni.lodz.pl; 3Department of Pharmacology and Therapeutics, School of Biomedical Sciences, College of Health Sciences, Makerere University, Kampala P.O. Box 7062, Uganda; jagodanowak@hotmail.com

**Keywords:** tanshinone, tanshinone derivative, ligand–protein interaction, signaling pathway

## Abstract

Tanshinones, biologically active diterpene compounds derived from *Salvia miltiorrhiza*, interact with specific proteins and DNA sequences, influencing signaling pathways in animals and humans. This study highlights tanshinone–protein interactions observed at concentrations achievable in vivo, ensuring greater physiological relevance compared to in vitro studies that often employ supraphysiological ligand levels. Experimental data suggest that while tanshinones interact with multiple proteomic targets, only a few enzymes are significantly affected at biologically relevant concentrations. This apparent paradox may be resolved by tanshinones’ ability to bind DNA and influence enzymes involved in gene expression or mRNA stability, such as RNA polymerase II and human antigen R protein. These interactions trigger secondary, widespread changes in gene expression, leading to complex proteomic alterations. Although the current understanding of tanshinone–protein interactions remains incomplete, this study provides a foundation for deciphering the molecular mechanisms underlying the therapeutic effects of *S. miltiorrhiza* diterpenes. Additionally, numerous tanshinone derivatives have been developed to enhance pharmacokinetic properties and biological activity. However, their safety profiles remain poorly characterized, limiting comprehensive insights into their medicinal potential. Further investigation is essential to fully elucidate the therapeutic and toxicological properties of both native and modified tanshinones.

## 1. Introduction

Tanshinones are a group of diterpenoid compounds characterized by a miltiradiene carbon skeleton. These compounds are synthesized and predominantly accumulated in the root periderm of the medicinal plant *Salvia miltiorrhiza* Bunge (*S. miltiorrhiza*), imparting its characteristic red–orange coloration [1,2,3,4]. The total tanshinone concentration in *S. miltiorrhiza* roots regenerated in vitro ranges from 0.269% to 1.137%, notably higher than levels reported in native plants growing in China (0.260–0.388%) or Poland (0.01–0.26%) [5,6,7]. These tanshinones constitute the lipophilic, biologically active fraction of dried *S. miltiorrhiza* roots, widely known as Danshen in China and Tanshen in Japan [8,9]. Analysis of crude *S. miltiorrhiza* extracts using high-speed counter-current chromatography (HSCCC) has identified seven major constituents: dihydrotanshinone I (DHTI, 8.2 mg), 1,2,15,16-tetrahydrotanshiquinone (5.8 mg), cryptotanshinone (CT, 26.3 mg), tanshinone I (TI, 16.2 mg), neo-przewaquinone A (25.6 mg), tanshinone IIA (TIIA, 68.8 mg), and miltirone (MT, 9.3 mg) from 400 mg of crude extract [10].

Although over 40 tanshinones have been identified, extensive biological investigations have focused on a few major compounds, such as CT, DHTI, TI, TIIA, and MT (Figure 1) [8,11,12,13,14,15]. These tanshinones have demonstrated diverse pharmacological activities, including the prevention of cardiovascular diseases (e.g., atherosclerosis, myocardial infarction, and ischemia/reperfusion injury) and anti-tumor properties [8,15,16,17,18]. The use of *S. miltiorrhiza* as a medicinal plant dates back to ancient China, where its therapeutic properties were first documented in the *ShennongBencao Jing* (Shennong’s Classic of Materia Medica), one of the oldest Chinese pharmacopeias [19]. However, the clinical application of tanshinones is hindered by their strong hydrophobicity and poor water solubility, leading to low oral bioavailability and suboptimal pharmacokinetic properties [17,20,21]. Common pharmacokinetic parameters, such as maximal plasma concentration (C_max_) and area under the curve (AUC), reflect limited systemic exposure following oral administration [17,21]. Recent strategies to enhance tanshinone bioavailability include the development of solid dispersions, β-cyclodextrin complexes, lipid nanoparticles, injectable formulations, and mixed ethanol–water extracts [22,23,24,25,26,27,28]. Additionally, synthetic tanshinone derivatives with improved water solubility and retained biological activity have been introduced to address these challenges (Figure 1) [29,30,31].

However, relatively little information is available on particular proteins directly interacting with tanshinones. While numerous studies have explored the molecular mechanisms underlying the medicinal properties of tanshinones, most focus on broader signaling pathways rather than specific protein targets. These studies often report altered expression of key pathway components or changes in overall signaling activity [32,33,34,35,36], representing significant progress in understanding tanshinone-mediated biological effects. However, data on direct protein–tanshinone interactions remain limited. Moreover, only a minority of these interactions have been characterized using experimentally validated and quantitative parameters, such as binding affinities (e.g., K_d_), inhibitory concentrations (IC_50_), or kinetic constants (K_i_) for enzymes [37,38,39]. These interactions serve as the initial triggers for downstream changes in signaling pathways, which ultimately drive the observed spectrum of clinical activities. Nevertheless, the precise mechanisms by which tanshinones exert their biological effects are not fully elucidated. Comprehensive and quantitative characterization of protein–tanshinone interactions across the human proteome is essential for a more detailed understanding of their biological functions and therapeutic potential.

The characterization of tanshinone–protein interactions often relies on computational techniques, including network pharmacology, structure-activity relationships (SAR), molecular docking, and data mining [40,41,42,43,44,45]. However, these findings frequently lack experimental validation, and studies performed in vitro often use tanshinone concentrations exceeding those achievable in vivo, raising concerns about their translational relevance [46,47,48].

This review aims to analyze the existing knowledge on interactions between tanshinones, their analogs, and biomolecules, including proteins and DNA. The data will be evaluated in the context of pharmacokinetic parameters observed following tanshinone administration, with a focus on identifying protein activities that are modulated not only under in vitro conditions but also in vivo.

Such analysis is expected to enhance our understanding of the initial tanshinone-induced alterations in signaling pathways, which have been extensively characterized in vitro. Furthermore, this approach could guide future research toward identifying novel metabolic pathways and biological processes influenced by tanshinones through precise characterization of their targets within the human proteome. Additionally, while bioinformatic approaches such as network pharmacology, molecular docking, and structure-activity relationship (SAR) studies have identified putative protein–tanshinone interactions, these predictions require validation through experimental and quantitative methods to deepen our understanding of the mechanisms underlying the clinical effects of tanshinones.

## 2. Selected Pharmacokinetic Parameters of Tanshinones

This study focuses on specific pharmacokinetic data for tanshinones, as summarized in Appendix A [21,22,25,26,27,49,50,51,52,53,54,55,56,57,58]. The analysis emphasizes the reported C_max_ values, which represent the maximal plasma concentrations of tanshinones and provide a direct basis for comparison with quantitative data on protein–tanshinone interactions. This approach allows for the evaluation of whether tanshinone concentrations used in in vitro or in vivo studies to induce protein–ligand interactions are consistent with pharmacokinetically achievable levels.

Assessing the alignment between experimental concentrations and observed C_max_ values enables a critical evaluation of the reliability of existing results, particularly regarding the ability of low tanshinone concentrations—within the range observed in pharmacokinetic studies—to elicit biological responses. Although C_max_ represents a transient peak in plasma concentration, it serves as a reasonable estimation of the concentrations that may be present in vivo following oral administration in humans and animals [17,21]. This comparison is crucial for understanding the translational relevance of in vitro findings to physiological conditions.

The C_max_ values, expressed in both µg/L and µM, were calculated based on the data in Appendix A and are summarized in Table 1. Two animal studies reporting significantly lower values were excluded from these calculations [26,53]. The available data indicate that the actual concentrations of tanshinones and miltirone observed in the blood following oral administration in animal studies range from 0.01 to 0.51 µM. Concentrations in the range of 0.20–0.40 µM appear to be relatively attainable in animal models after oral dosing.

Tanshinone formulations designed to enhance bioavailability generally do not significantly alter the observed concentration range, which remains within 0.08–0.56 µM (Table 1). Studies in humans reveal that typical tanshinone concentrations in the blood following oral administration are notably low, ranging from 0.002 to 0.02 µM. However, micronized *S. miltiorrhiza* preparations can increase these concentrations to 0.02–0.49 µM (Table 1). Substantially higher C_max_ values are achieved with intravenous administration. For example, intravenous administration of sodium tanshinone IIA sulfonate (STS) yields a C_max_ of 1.88 µM (Table 1) [58].

## 3. Indirect Effects of Tanshinone Interaction with Proteins and DNA

Certain tanshinones directly inhibit enzymes, disrupting metabolic pathways or altering metabolite levels [38,45,46,47,48]. Additionally, they may modulate signaling pathway activity, exerting broader effects by influencing the expression of numerous genes [30,31,32,33,34,54]. A key factor contributing to the wide-ranging, indirect effects of tanshinones on metabolic pathways and gene expression is their ability to interact with DNA, as well as with proteins directly associated with DNA and RNA [59,60,61,62,63,64,65].

DHTI can influence RNA levels by interacting with the RNA-binding protein HuR (human antigen R), which recognizes U/AU-rich sequences in RNA using its two RNA-recognition motifs, RRM1 and RRM2 [60].

Messenger RNAs with a higher abundance of AU-rich regions show stronger affinity for HuR, leading to more frequent binding and higher representation. By interacting near the linker between the RRM domains, DHTI stabilizes HuR in a closed state. This interaction prevents HuR from binding to RNAs with a lower density of AU-rich regions while enhancing its binding to RNAs with a higher density. These effects occur because DHTI competes with RNA for the same binding sites on HuR, as demonstrated by RNA mobility shift and alpha screen experiments.

Further studies using ribonucleoprotein immunoprecipitation and microarray analysis in HeLa cells revealed that DHTI modifies HuR’s RNA interactions. Among 2306 HuR-associated mRNAs, DHTI decreased binding to 79 and increased the levels of 558 mRNAs [60]. The binding strength of DHTI to HuR is significant, with an in vitro dissociation constant of 50 nM, which is sufficient to inhibit 50% of HuR’s interaction with a biotinylated single-strand AU-rich RNA probe [61]. Synthetic DHTI derivatives with enhanced binding affinities (as low as 12.8 nM and 15 nM) have also been developed, showing greater potency than DHTI itself (Figure 2). Achieving these nanomolar concentrations of DHTI in the bloodstream is possible under in vivo conditions following oral administration of tanshinones (Table 1).

Research on TIIA suggests it binds to the DNA minor groove, with a preference for AT-rich sequences. The strength of this interaction is significantly influenced by the structural features of TIIA, including the oxygen in the furan ring, two hydroxyl groups on ring A, and the ortho-quinone moiety in ring C. These features enable the formation of strong hydrogen bonds with DNA, enhancing the DNA–ligand interaction and contributing to its pronounced cytotoxic effects [62]. TIIA binding induces conformational changes in DNA, characterized by a decrease in the positive band and an increase in the negative band. These structural alterations correlate with a reduced occurrence of RNA polymerase II (RNAPII) at the P2 promoter, intron, and exon regions of the *c-myc* oncogene, as observed in vitro. Notably, this reduction in RNAPII binding was evident at a TIIA concentration as low as 0.2 µM, which is readily achievable in vivo (Table 1) [63]. Higher concentrations of TIIA, ranging from 4 to 20 µM, were required to induce RNAPII phosphorylation and degradation, followed by p53 activation and apoptosis. However, such concentrations are not typically observed under in vivo conditions [63].

The synthetic derivative of TIIA, 1,6,6-trimethyl-11-phenyl-7,8,9,10-tetrahydro-6*H*-furo[2′,3′:1,2]phenanthro[3,4-d]imidazole (TA25), binds to the G-quartet plane formed by G7, G11, G16, and G20 within the *c-myc* G-quadruplex DNA through π-π stacking interactions [64]. This binding mechanism inhibits the proliferation of lung cancer A549 cells, with an IC_50_ value of 17.9 μM [64]. The same G-quadruplex structure is present in the promoters of genes such as *K-ras*, *VEGF*, *Bcl-2*, and *Tel-26*. Interactions with 2-phenyl-1*H*-imidazole-based tanshinone derivatives downregulate the expression of these genes, leading to effects such as S-phase arrest in breast cancer MDA-MB-231 cells, a reduction in tumor growth rates, and fewer metastatic cells (Figure 2) [65].

Among eight tested inhibitors, the most effective compounds exhibited mean IC_50_ values of 12.8 μM and 15.2 μM against MDA-MB-231 and HepG2 cell lines, respectively. However, these concentrations are not achievable in vivo under typical blood plasma conditions (Table 1) [65].

An alternative explanation for the relatively broad and indirect effects on gene expression following TI administration could involve its interaction with components of the Polycomb repressive complex 2 (PRC2), including zebrafish EZH2 (zEZH2), human EED (hEED), and zSUZ12 [59].

However, the interactions between TI and these components—zEZH2, hEED, and zSUZ12a and b—are relatively weak, with dissociation constants (K_D_) of 23.967 μM, 94.48 μM, 1518 μM, and 762 μM, respectively. Despite these weak interactions, TI has been shown to inhibit PRC2-mediated methylation of H3K27 in vitro, which could lead to the derepression of certain genes [66].

Moreover, a concentration of 5 µM is required for these effects to take place, which is significantly higher than the levels typically observed in vivo [59].

The PRC2 complex consists of an enhancer of zeste homolog 2 (EZH2) or enhancer of zeste homolog 1 (EZH1), embryonic ectoderm development (EED), and suppressor of zeste 12 protein homolog (SUZ12) [67,68,69,70]. EZH2 functions as a histone methyltransferase, interacting with EED and SUZ12 to catalyze the transfer of a methyl group from the cofactor S-adenosylmethionine (SAM) to the ε-amino group of lysine residues [67,68,69,70]. Additionally, EED stabilizes the PRC2 complex, enhances its histone methyltransferase activity, and promotes chromatin expansion through trimethylation of H3K27, driven by EED–EZH2 interactions. EED further stabilizes the active site of PRC2 and provides a docking platform for factors that assemble the PRC2 complex, while SUZ12 interacts with EZH2 [67,68,69,70]. The components of the PRC2 complex are conserved across eukaryotes, suggesting the potential inhibitory activity of TI against human homologs of EZH2 and SUZ12a/b [71,72,73].

One possible reason for the indirect effects of tanshinones on metabolic pathways or gene expression is the ability of TIIA to specifically block the redox activity of human apurinic/apyrimidinic endonuclease 1/redox factor-1 (APE1/Ref-1) while preserving its DNA repair functions [74]. Inhibition of APE1/Ref-1 protects against the redox modification of key transcription factors such as nuclear factor-κB (NF-κB), activator protein 1 (AP-1), and hypoxia-inducible transcription factor 1α (HIF-1α), preventing their interaction with DNA [74]. Notably, a significant decrease in the binding of NF-κB, AP-1, and HIF-1α to DNA was observed following administration of TIIA at a concentration of 8 µM. While the dissociation constant (Kd) for the interaction between APE1/Ref-1 and TIIA, measured by dual polarization interferometry, is only 0.88 nM, the IC_50_ values for HeLa cell proliferation were 29.6 µM, 12.1 µM, and 4.8 µM after 24, 48, and 72 h of treatment, respectively. Given that much lower concentrations of TIIA are achievable in vivo, it is challenging to observe these effects in living organisms [74].

In addition, tanshindiols B and C were found to inhibit the methyltransferase activity of EZH2 in an in vitro enzymatic assay, with IC_50_ values of 0.52 µM and 0.55 µM, respectively [75].

## 4. Direct Effects of Tanshinone Interaction with Proteins

The effects of enzyme inhibition following tanshinone administration are generally more direct compared to the previously discussed indirect interactions [37,38,45,48,49]. In these cases, the reduction in the concentration of a metabolite produced by the inhibited enzyme offers a clearer and more straightforward explanation of the biological and clinical outcomes.

However, enzymes involved in key metabolic processes, such as glycolysis (e.g., enolase), the citric acid cycle, and oxidative phosphorylation (e.g., succinate dehydrogenase), when inhibited by tanshinones, can lead to significant and broad changes in signaling pathways [76,77].

### 4.1. Tanshinone–Protein Interactions in Oncology

In the context of cancer treatment, enzymes involved in tryptophan catabolism, particularly those converting tryptophan to kynurenine, are important targets for tanshinones. Elevated levels of kynurenine and an increased kynurenine/tryptophan (Kyn/Trp) ratio have been observed in patients with advanced cancers, where tumors evade immune system control [78,79].

This shift contributes to an immunosuppressive phenotype, driven by the agonistic activity of kynurenine on aryl hydrocarbon receptors [80,81]. As a result, inhibiting enzymes such as indoleamine 2,3-dioxygenase 1 (IDO1) and tryptophan 2,3-dioxygenase (TDO) could reduce tryptophan catabolism, lower kynurenine concentrations, and modulate the Trp-kynurenine axis, thereby restoring anti-tumor immunity and improving the body’s ability to fight cancer [82].

Among small molecules tested as potential inhibitors of IDO1 and TDO, compounds such as CT and TIIA have been evaluated, with their mean IC_50_ concentration values ranging from 9.8 to 25.1 µM [83]. At these concentrations, their ability to influence the Trp-Kyn pathway in vivo, given the typical blood concentrations, is unlikely. However, synthetic tanshinone derivatives, such as 3(S),17-dihydroxytanshinone, demonstrated much lower IC_50_ values, ranging from 0.59 to 1.01 µM, indicating stronger inhibitory activity [83].

Even more promising results were observed for analogs containing fluorine, chlorine, or acetoxyl at the C-17 methyl group position, combined with a 3-keto group (Figure 3A). These derivatives exhibited a mean IC_50_ values range of 0.07 to 0.42 µM against IDO1 and 0.11 to 0.57 µM against TDO. Additional modifications, such as oxidation of the 3-hydroxyl group to a 3-keto group or acetylation, further enhanced inhibitory activity against both IDO1 and TDO. This effect was particularly pronounced when the C-17 methyl group was substituted with hydrogen or acetoxyl, along with a hydroxyl group. The mean IC_50_ values for these analogs against IDO1 ranged from 0.16 to 1.73 µM, and against TDO, they ranged from 0.20 to 0.67 µM (Figure 3B) [83].

Unfortunately, the precise bioavailability, pharmacokinetic parameters (such as C_max_), and potential adverse effects of these synthetic tanshinone derivatives remain unknown [83].

However, another synthetic derivative with broad clinical applications and a well-studied safety profile, STS, has been shown to inhibit both IDO1 and a variant of TDO, known as TDO2. While the IC_50_ value for IDO1 was found to be 10 µM—well above the concentrations achievable in vivo—the IC_50_ value for TDO2 is only 1 µM, suggesting that STS may have the potential to reduce TDO2 activity under in vivo conditions [84].

In addition, four native tanshinones were found to have weak inhibitory activity against IDO1, with inhibition rates ranging from 5.6% to 78.8% at 20 µM. However, a tanshinone derivative identified in a natural products library of 2000 compounds exhibited much stronger inhibitory activity, with IC_50_ values of 2.8 µM against IDO1 and 5.1 µM against TDO2, respectively [47].

Tanshinones have been shown to induce apoptosis in the HCT116 tumor cell line by targeting diadenosine triphosphate (Ap3A) hydrolase, a member of the human fragile histidine triad (FHIT) protein family. Native tanshinones such as TIIA, TI, and isocryptotanshinone are relatively weak inhibitors of Ap3A, with mean IC_50_ values ranging from 4 to 6 µM, while STS exhibits a stronger inhibitory effect with an IC_50_ value of 2.2 ± 0.05 µM [85].

Similarly, weak inhibitory activity has been observed for protein kinase C and fatty acid synthase, which is not significant at concentrations typically achievable in vivo. While TIIA inhibits protein kinase C activity, only a 42% reduction in activity is observed at a concentration of 20 µM [48]. Additionally, compounds such as 15,16-dihydrotanshinone I, CT, TI, and TIIA inhibit fatty acid synthase activity by 50% (IC_50_) at concentrations ranging from 12.0 to 30.3 µM [37].

Tanshinone I and its two derivatives, S222 and S439, directly inhibit DNA topoisomerase I and II (Top1/2) (Figure 4) [86]. All three compounds induce DNA double-strand breaks, G2/M cell cycle arrest, and apoptosis, either in a p53-dependent or p53-independent manner, with the response varying based on the specific compound used [86].

Due to significantly improved water solubility, S222 and S439 demonstrated approximately 12- and 14-fold more potent inhibitory effects on cell proliferation compared to TI in a panel of 15 cancer cell lines. The mean IC_50_ values for S439 and S222 were 1.56 µM (range: 0.67–4.51 µM) and 1.79 µM (range: 0.53–6.11 µM), respectively [86].

These differences were also reflected in the concentrations required for supercoiled DNA relaxation, with TI requiring 10 µM, while S439 and S222 required only 1 µM each. This was measured by the accumulation of γH2AX, a known biomarker for double-strand breaks, after up to 12 h of incubation [86]. Both derivatives showed more potent inhibition of Top2 than the classical Top2 inhibitor etoposide (VP-16). Furthermore, TI and its derivatives differ structurally from known Top1, Top2, and dual Top1/2 inhibitors [87,88,89].

The mechanisms of action for S222 and S439 also vary depending on the p53 status of the cells. S222 induced both G2/M arrest and apoptosis in a p53-independent manner, while S439 induced G2/M arrest only in p53-proficient cells but led to significantly more apoptosis in p53-deficient cells compared to p53-proficient ones [86].

Recently developed dihydrofuran-based derivatives of TI exhibited improved cytotoxicity against KB and drug-resistant KB/VCR cancer cell lines, with mean proliferation IC_50_ values ranging from 1.24 to 2.62 µM, compared to native TI, which showed IC_50_ values of 4.40 to 5.87 µM. However, another compound, in which the C-7 position was substituted with an N-3-(diethylamino)propyl group, emerged as an exceptionally potent inhibitor of KB and KB/VCR cell proliferation, with IC_50_ values ranging from 0.12 to 0.33 µM. This compound also demonstrated much better water solubility (15.7 mg/mL), and oral bioavailability (21%) compared to native TI [90].

Among a group of other tanshinone analogs produced by the condensation of furophenanthraquinones, two compounds—naphtho[1,2-b]thiophene-4,5-dione and naphtho[2,1-b]thiophene-4,5-dione—showed approximately four-fold improved anti-proliferative activity, with IC_50_ values of 1.86–1.95 µM for MCF7 and 2.60–2.96 µM for MDA-MB-231 cancer cell lines, compared to natural tanshinones [91].

The introduction of a methoxy group at the C8 position of TIIA led to a derivative that binds to the colchicine site on tubulin, inhibiting its assembly and disrupting normal microtubule network formation. This derivative showed a mean proliferation rate IC_50_ values ranging from 0.28 to 3.16 µM across six tested cell lines: HepG2, HeLa, MCF-7, MGC, A549, and U937 [92].

Recently developed analogs of TIIA, produced by the addition of a substituted benzene group through the Debus–Radziszewski reaction, resulted in a series of compounds demonstrating strong cytotoxic effects against two prostate cancer cell lines, LNCaP and CWR22Rv1 [93]. A particularly high cytotoxic activity was observed in a derivative containing a carbonyl group and three methoxy groups in the benzene ring. The mean IC_50_ values for growth inhibition of LNCaP and CWR22Rv1 cell lines were 0.41 µM and 0.74 µM, respectively, approximately 20-fold and 19-fold lower values compared to TIIA. Moreover, the IC_50_ value of this prepared compound was comparable to the activity of enzalutamide, a well-known drug used in the treatment of castration-resistant prostate cancer, which showed IC_50_ values of 0.59 µM for LNCaP and 0.83 µM for CWR22Rv1 [93].

### 4.2. Tanshinone–Protein Interactions Modify the Drug Metabolism

Tanshinones and MT modulate the activity of enzymes involved in both the first and second phases of drug metabolism [94,95,96,97]. In the first phase, the majority of protein–tanshinone interactions are associated with cytochrome P450 (CYP) enzymes [38,96,97,98,99]. However, these interactions are generally weak, necessitating ligand concentrations that are considerably higher than those observed in vivo to achieve the reported IC_50_ values [38,96].

DHTI demonstrated weak inhibition of CYP2D6 activity, with an IC_50_ value of 35.4 µM, as determined by the O-demethylation of dextromethorphan to dextrorphan in human pooled microsomes. In comparison, the inhibition observed for three other tested tanshinones was even weaker, with an IC of approximately 20% at significantly higher ligand concentrations ranging from 16.5 to 61.4 µM [96].

Among the tested compounds was MT, a precursor in the biosynthesis of tanshinones in plants, known for its antioxidative, anxiolytic, and anticancer effects [10,11,12,100,101]. MT exhibited moderate inhibition of CYP1A2 (IC_50_ 1.73 µM) and CYP2C9 (IC_50_ 8.61 µM), along with weak inhibition of CYP2D6 (IC_50_ 30.20 µM) and CYP3A4 (IC_50_ 33.88 µM) [94].

Moreover, the in silico data suggest that the MT binds more strongly to CYP2D6 than fluoxetine, and Phe120 may be the key aa residue, leading to the inhibition of CYP2D6-mediated fluoxetine N-demethylation. Although the IC_50_ value for CYP2D6.1 inhibition by MT was 2.90 µM, the experimental results proved, that after the per os administration of MT (40 mg/kg), the concentration of fluoxetine in rats increased markedly [96]. This increase in fluoxetine levels could be attributed to the potential simultaneous inhibition of other CYP enzymes [57,102]. This hypothesis is supported by the fact that after oral administration of 40 mg/kg miltirone to rats, the MT concentration reached approximately 92.7 µg/L (0.33 µM), which is much lower than the IC_50_ value for CYP2D6.1 inhibition (2.90 µM) [57,102].

Inhibition of the ω-hydroxylation of arachidonic acid (AA) by CYP4A11, CYP4F2, and CYP4F3B may reduce the concentration of proangiogenic and mitogenic 20-hydroxyeicosatetraenoic acid (20-HETE), thereby modulating cerebral ischemia/reperfusion (I/R) injury and tumor promotion processes [103]. Four primary tanshinones were found to be weak inhibitors of arachidonic acid (AA) ω-hydroxylation by recombinant CYP4A11, CYP4F2, and CYP4F3B. TIIA exhibited noncompetitive inhibition of CYP4F3B, with a Ki value of 4.98 µM. Additionally, the CT (Ki 6.87 µM), TI (Ki 0.42 µM), and DHTI (Ki 0.09 µM) acted as mixed-type inhibitors of CYP4A11. DHTI showed noncompetitive inhibition of CYP4F2 (Ki 4.25 µM) and CYP4F3B (Ki 3.08 µM). Further investigations revealed that 20-HETE formation was significantly inhibited by high concentrations of DHTI (5 and 20 µM), concentrations that exceed those typically observed in vivo [103].

A related study demonstrated that TIIA and TI are moderate competitive inhibitors of CYP2C8, with Ki values of 1.18 µM and 4.20 µM, respectively. DHTI was identified as a moderate noncompetitive inhibitor of CYP2J2 (Ki 6.59 µM) but exhibited stronger inhibitory activity against CYP2C8, with a Ki value of 0.43 µM [98].

Although the concentrations of tanshinones observed in the blood are not sufficient to significantly inhibit the activity of various CYP enzymes, they may still influence the pre-systemic metabolism of different drugs [98,103,104,105]. These effects are more likely to occur due to the considerably higher concentrations of tanshinones in the small intestine compared to the blood [106].

In studies involving rats orally administered TIIA at a dose of 60 mg/kg, the mean time-dependent (0.5–24 h) TIIA concentration in the blood ranged from 1.88 to 6.08 µg/L (0.006–0.021 µM). In contrast, under the same conditions, the mean TIIA concentration in the small intestine was approximately five hundred times higher, ranging from 1020 to 2885 µg/L, corresponding to 3.47–9.80 µM. The concentration in the liver was also significantly higher compared to the blood, ranging from 0.53 to 1.57 µM. However, the highest concentration of TIIA was observed in the stomach, where it reached a maximum of 9.52–48.15 µM, levels not seen in other organs [106].

UDP-glucuronosyltransferases (UGTs) are key enzymes involved in phase II drug metabolism, responsible for catalyzing the glucuronidation reaction, which involves the addition of a glucuronosyl group from UDP-glucuronic acid (UDPGA) to the substrate [107]. The activity and function of these enzymes can also be modulated by tanshinones [108,109]. CT inhibited UGT1A7 and UGT1A9 with IC_50_ values of 1.91 ± 0.27 µM and 0.27 ± 0.03 µM, respectively, while DHTI inactivated UGT1A9 with an IC_50_ value of 0.72 ± 0.04 µM [108].

Research by Liu et al. (2022) demonstrated that TI (0–1 µM) inhibits the activity of human UGT1A3, 1A6, and 1A7, as well as rat UGT1A3, 1A6, 1A7, and 1A8. Additionally, CT (0–1 µM) inhibits the activity of human UGT1A3 and 1A7, and rat UGT1A7, 1A8, and 1A9 [109].

Recently, tanshinones and their analogs have been identified as potent inhibitors of human carboxylesterases (hCE), with the ability to modulate the biological activity of esterified drugs [110,111,112]. Research on *S. miltiorrhiza* extract fractions revealed the presence of up to 7 mg/g of anhydrides of TIIA and CT [111]. These components act as extremely potent and irreversible inhibitors of hCE1 and human intestinal carboxylesterase (hiCE), with Ki values below 1 nM [111]. The strength and stability of inhibition are mediated by the interaction with the hydroxyl group of Ser221 in the enzyme’s active site, forming a stable ester derivative. A significant increase in biological inhibitory activity against hCE1 and hiCE was observed for all four tested tanshinone–anhydride pairs, with inhibitory potency rising by several thousand-fold. The most pronounced increase in inhibitory activity was noted for the TI anhydride. While the mean Ki for hCE1 and hiCE inhibition by TI was 26,250 nM (26.25 µM) and 14,550 nM (14.55 µM), respectively, the TI anhydride exhibited mean Ki values of 2.40 nM and 0.82 nM for hCE1 and hiCE, reflecting a 10,938-fold and 17,744-fold increase in inhibition potency, respectively [111].

Other tanshinone derivatives containing a 1,2-dione group as part of a naphthoquinone core were synthesized through a novel and more efficient route, which involved Suzuki coupling followed by electrocyclization and oxidation of o-phenanthroquinones. Among the resulting compounds, several exhibited strong inhibitory activity against hCE1 and hiCE, with mean Ki values against these enzymes being below 15 nM [112].

### 4.3. Tanshinone–Protein Interactions in Biological Processes and Diseases

#### 4.3.1. Neurodegenerative Diseases and Pain Treatment

Monoamine oxidase B (MAO-B), acetylcholinesterase (AChE), and butyrylcholinesterase (BChE) are important targets for drugs used in the treatment of Alzheimer’s disease [113,114,115]. Tanshinones and their derivatives interact with these enzymes and inhibit their activity. However, the inhibition measured by IC_50_ values is relatively weak and not easily reproducible at concentrations typically achievable in vivo [116,117,118,119]. TI, TIIA, and CT have been shown to inhibit human MAO-A with IC_50_ values of less than 10 µM, but they are much weaker inhibitors of hMAO-B activity, with IC_50_ values above 25 µM [116].

Results from Hatfield et al. (2018) demonstrated that at a concentration of 10 µM, TI, TIIA, CT, DHT, and their anhydrides inhibited AChE and BChE activity by 1–84% [111]. Inhibition of AChE was also observed for 15,16-dihydrotanshinone (15,16-DHT), which moderately inhibited AChE activity by 65.17 ± 1.39% at 10 µM [112]. DHTI exhibited weak inhibition of both AChE and BChE, with IC_50_ values of 1.50 ± 0.02 µg/mL (5.39 ± 0.07 µM) and 0.50 ± 0.01 µg/mL (1.80 ± 0.04 µM), respectively [118].

Derivatives of tanshinones are also weak inhibitors of AChE and BChE [117,119]. Among the tested compounds, 1,2-didehydromiltirone exhibited an IC_50_ value of 1.12 ± 0.07 μg/mL (3.99 ± 0.25 µM), cryptotanshinone CT had an IC_50_ of 1.15 ± 0.07 μg/mL (3.88 ± 0.24 µM), 1,2-didehydrotanshinone IIA showed an IC_50_ value of 5.98 ± 0.49 μg/mL (20.46 ± 1.68 µM), and 1(S)-hydroxytanshinone IIA demonstrated an IC_50_ value of 5.71 ± 0.27 μg/mL (18.40 ± 0.87 µM) [119]. Additionally, (1*R*,15*R*)-1-acetoxycryptotanshinone and (1*R*)-1-acetoxytanshinone IIA, which were isolated from *Perovskia atriplicifolia*, selectively inhibited BChE, with mean IC_50_ values of 2.4 µM and 7.9 µM, respectively (Figure 5) [117].

Deoxyneocryptotanshinone is a mixed-type inhibitor of beta-secretase, also known as β-site amyloid precursor protein cleaving enzyme 1 (BACE1), with an IC_50_ value of 11.53 ± 1.13 µM [120]. Inhibitors of BACE1 are considered potential therapeutic agents for Alzheimer’s disease [121].

Tanshinone IIA (TIIA) inhibits human monoacylglycerol lipase (MAGL), a critical hydrolase in the endocannabinoid system, with an IC_50_ value of 0.26 µM. MAGL has been identified as a potential target for pain treatment. Among TIIA derivatives, the most potent compounds are *N*-methyl-heterocyclicacetamideanalogs substituted at C-17, with the lowest IC_50_ value of 0.12 µM. One of the selected compounds exhibited significantly improved water solubility compared to TIIA (417.3 µg/mL (1.42 mM) vs. 138 µg/mL (0.47 mM) and demonstrated 72.5% of the analgesic activity of morphine at a dose of 5 mg/kg administered subcutaneously in an acetic acid-induced writhing model [122].

#### 4.3.2. Blood Clotting and Fibrinolysis

Tanshinones influence blood clotting and fibrinolysis, although at concentrations that significantly exceed those typically observed in vivo [123,124,125]. CT inhibits the activity of microsomal prostaglandin E synthase-1 (mPGES-1) and 5-lipoxygenase (5-LO), with mean IC_50_ values of 1.9 µM and 7.1 µM for mPGES-1 and 5-LO, respectively [123].

Prostaglandin E plays a role in platelet activation, while leukotrienes produced by 5-LO are involved in inflammation processes [126,127,128,129]. In contrast, TIIA does not exhibit significant inhibition of either enzyme [123]. The interaction of CT with mPGES-1 and 5-LO affects clotting parameters, leading to an increased bleeding time in mice (2.44 ± 0.13 min) and modulating clot retraction (0.048 ± 0.011) [123].

TI and TIIA are potential inhibitors of factor Xa, with mean IC_50_ values of 112.59 µM and 138.19 µM, respectively [124]. Additionally, STS and CT, inhibit plasminogen activator inhibitor-1 (PAI-1), the primary inhibitor of both tissue-type plasminogen activator (tPA) and urokinase-type plasminogen activator (uPA). PAI-1 plays a critical role in regulating the fibrinolytic system [125,130]. The IC_50_ values for STS and CT in the uPA/PAI-1 assay are 41 µM and 98 µM, respectively, while in the tPA/PAI-1 assay, the IC_50_ values are 59 µM and 152 µM, respectively [125].

Additionally, treatment with STS, administered intravenously at a dose of 60 mg/day for 10 days, has been shown to reduce blood–brain barrier damage and improve therapeutic outcomes in acute ischemic stroke patients treated with recombinant tissue plasminogen activator (rt-PA) [131].

#### 4.3.3. Bone Metabolism and Osteoporosis

Tanshinones inhibit the collagenase activity of cathepsin K (CatK) by preventing CatK oligomerization [132,133]. This inhibition occurs without interfering with the CatK active site, offering a potential alternative treatment for osteoporosis that avoids the side effects typically associated with active site-directed inhibitors [134]. These novel inhibitors, which target exosites rather than allosteric sites, are referred to as ectosteric inhibitors, distinguishing them from traditional allosteric inhibitors [133,134,135]. Among the tested tanshinones and their derivatives, the soluble and insoluble collagen type I degradation activities were most notably inhibited by STS sodium salt, with IC_50_ values of 2.7 ± 0.2 µM and 3.6 ± 1.7 µM, respectively [133].

In contrast to the active site inhibitor odanacatib, STS does not affect the metabolic activity or osteoclastogenesis of osteoclasts (Ocs) and effectively suppresses bone resorption in both human (IC_50_ 237 ± 60 nM) and mouse (IC_50_ 245 ± 55 nM) osteoclasts [134].

When applied at concentrations corresponding to their IC_50_ values, STS (245 nM) and odanacatib (15 nM) both suppress bone-resorbing activity in human Ocs. At these inhibitor concentrations, osteoclasts were still able to form short trenches, but their activity was frequently interrupted. However, when the inhibitor concentrations were increased to 2 to 5 times the IC_50_ values, osteoclasts lost the ability to form trenches and were restricted to forming only pits [136].

Tanshinones may improve bone metabolism and protect against osteoporosis by inhibiting the biotransformation and clearance of estrogen, leading to increased estrogen concentrations [137,138]. The mean IC_50_ values for TIIA (0.65 µM) and 15,16-dihydrotanshinone (51.60 µM) in inhibiting sulfotransferase SULT1A1 suggest that TIIA has the potential to affect estrogen sulfation [137].

#### 4.3.4. Diabetes and Obesity

Tanshinones could reduce postprandial glucose and insulin levels by inactivating the α –glucosidase, an enzyme that breaks down carbohydrates, situated in the intestine epithelial wall [139]. IC_50_ value of α-glucosidase inhibition by TIIA is 11.39 ± 0.77 µM. However, the twelve TIIA derivatives of IC_50_ values from 0.73 ± 0.11 to 9.46 ± 0.57 µM were proposed for putative treatment of type 2 diabetes mellitus [140].

Eight naturally occurring tanshinones and tanshinone derivatives have been identified as potent inhibitors of 11β-hydroxysteroid dehydrogenase 1 (11β-HSD1), with IC_50_ values in the nanomolar range for both human (0.5–206.5 nM) and mouse (0.4–392.3 nM) models. These compounds also demonstrate good selectivity for 11β-HSD1 over 11β-HSD2. 11β-HSD1 is considered a promising target for the treatment of metabolic syndrome, including conditions such as obesity and type 2 diabetes. Further development of synthetic tanshinone derivatives has resulted in a group of 11β-HSD1 inhibitors with IC_50_ values in the nanomolar range in in vitro experiments and between 1.4 to >4.0 µM in in vivo tests on 3T3-L1 adipocytes [141].

CT and TIIA are potential pancreatic lipase (PL) inhibitors, with CT exhibiting stronger lipase inhibitory activity (IC_50_ 6.86 ± 0.43 µM) compared to TIIA. This suggests that CT could serve as a lead compound for the development of more effective PL inhibitors [142]. Since PL hydrolyzes 50–70% of dietary triacylglycerol, its inhibition could decrease the lipid digestion rate, potentially offering benefits in the treatment of obesity [142,143].

#### 4.3.5. SARS-CoV-2 Treatment

Derivatives of STS were evaluated as potential inhibitors of the SARS-CoV-2 papain-like protease (PLpro), but their IC_50_ values were found to be quite high, ranging from 68.3 µM to over 100 µM [144].

Among the tested tanshinones, TI demonstrated the most potent inhibitory activity at the nanomolar level against the deubiquitinating activity of the SARS-CoVPLpro viral cysteine protease, with an IC_50_ value of 0.7 µM [145].

#### 4.3.6. Other Potential Applications of Tanshinone–Protein Interactions

TI has been identified as an inhibitor of spleen tyrosine kinase (Syk), exhibiting an average IC_50_ value of 1.64 µM. Additionally, it demonstrated anti-mast cell degranulation activity in vitro, with a mean IC_50_ value of 2.76 µM. The inhibition of Syk-mediated activation of immunoreceptors impedes the activation of mast cells, macrophages, and B cells, thereby preventing the release of inflammatory mediators. This mechanism positions Syk as a promising therapeutic target for the treatment of autoimmune and inflammatory disorders [146,147].

TIIA has been identified as a tyrosinase inhibitor, with a mean IC_50_ value of 1214 µM (1.214 mM). This suggests its potential use in the development of therapeutic agents aimed at locally treating conditions associated with excessive melanin production, such as melasma, freckles, and melanosis [148]. Additionally, STS demonstrated the ability to inhibit L-type Ca^2+^ channels, with an EC_50_ value of 59.5 µM, leading to vasorelaxation in rat mesenteric arteries. These findings highlight the potential of STS in modulating vascular tone and influencing related physiological processes [149].

## 5. Tanshinone-Dependent Regulation of Signaling in the In Vivo Conditions

The presented analysis indicates that instances of enzyme-tanshinone interactions resulting in a significant reduction of enzyme activity (50% or greater) at concentrations achievable in vivo in human or animal blood are infrequent. Such interactions were observed only in a limited number of cases, as outlined in Table 1 and Table 2. The obtained information does not account for the organ-specific distribution of tanshinones, which may reach significantly higher concentrations in localized tissues such as the intestine, liver, or stomach compared to the bloodstream. This disparity suggests that tanshinones could potentially inhibit a broader range of enzymes, with IC_50_ values ranging from 0.53 to 48.15 µM, as reported in previous studies [106].

The limited number of well-established tanshinone–protein interactions observed in in vivo conditions contrasts with the considerably larger number of signaling pathways influenced by tanshinones and their derivatives in animal and human studies (Table 2 and Appendix A) [45,150,151,152,153,154,155,156,157,158,159,160,161,162,163,164]. Among the affected signaling pathways are several, leading to the activation of NF-κB, androgen receptor, NEAT1, and cJUN, which modulate the expression of multiple genes [45,150,158,159,163,164].

## 6. Structure-Activity Relationship of Tanshinones and Their Derivatives

The anti-cancer effects of TI, TIIA, and six derivatives of TIIA on normal and cancerous colon cells have been analyzed, revealing that the naphthalene or tetrahydronaphthalene structures in rings A and B, along with the ortho-quinone moiety in ring C, are crucial for the bioactivity of tanshinones. Structural modifications in ring A and alterations to the furan or dihydrofuran groups in ring D were found to significantly influence activity [165]. Enhanced cytotoxicity was observed with increased delocalization within rings A and B, while a non-planar and compact D ring conferred improved anti-cancer activity. Structure–activity relationship (SAR) studies further indicated that the presence of polar and electron-withdrawing groups, such as -F, -NO_2_, -OH, or -CF_3_, at the para position of aromatic aldehydes significantly enhanced the activity. Conversely, substituting a -Br group in the furan ring of TIIA abolished its anti-cancer properties [165].

The findings align with those of Zhao et al. (2011), which demonstrated enhanced cytotoxicity of tanshinones following the introduction of polar substituents into the A or D rings [166]. The critical importance of an intact D ring, especially in its unsaturated form, was confirmed through cytotoxicity studies on H1299 and Bel-7402 cell lines. Furthermore, the presence of a 3-OH group has been identified as playing a significant role in the anti-cancer activity of tanshinones [167].

Structure–activity relationship (SAR) studies provide insights into the variations in antioxidant activity of tanshinones, which are mediated by the activation of nuclear factor erythroid 2–related factor 2 (Nrf2) [168]. A key mechanism for Nrf2 activation by quinones involves electrophilic modification of cysteine residues in Keap1, a critical negative regulator of Nrf2 (Abiko). Consequently, the higher electrophilic nature of tanshinones enhances their electron-abstracting capacity, leading to a greater potential to activate Nrf2 and stronger antioxidant properties [168,169].

Two parameters were employed to assess the electrophilic properties of tanshinones, namely electron affinity and the energy level of the lowest unoccupied molecular orbital (LUMO), in order to describe the indirect antioxidant activity of tanshinones and their derivatives [168]. Both parameters indicate that TI and DHTI exhibit stronger electronacceptor properties compared to TIIA andCT. The observed differences between these two groups of tanshinones are attributed to variations in the structure of ring A, with a benzene ring present in TI and DHTI, and a cyclohexane ring in TIIA and CT [168]. Based on the Hammett constant, the electron-donating ability of methyl groups derived from cyclohexane is stronger than that of a benzene ring. Consequently, the more electrophilic properties of benzene-containing tanshinones may account for their enhanced potential to activate Nrf2. This mechanistic explanation also applies to the distinct structural differences between TI, TIIA, DHT, and CT. TI and TIIA feature a double bond in ring D, whereas DHT and CT possess only single bonds. According to the Hammett constant values, the methyl group is a stronger electron-donating group compared to the ethylene group. As a result, TI and TIIA exhibit greater activity than DHT and CT in electron abstraction, leading to subtle differences in Nrf2 activation and subsequent antioxidant properties [168].

Structural studies of tanshinones as agonists of the human estrogen receptor α ligand-binding domain (hERα-LBD) revealed that the binding affinity of tanshinones for hERα-LBD increases with their Connolly solvent-excluded volume (CSEV) [170]. Cryptotanshinone, having the largest volume, exhibits the strongest binding to the receptor, while tanshinone I, with the smallest volume, shows the weakest binding affinity for hERα-LBD. It is hypothesized that the larger compounds facilitate stronger hydrophobic interactions compared to smaller molecules [170].

Other structure–activity relationship (SAR) studies on tanshinones by Wang et al. (2011) demonstrated that the presence of a double bond at position 15 of the furan ring is associated with the competitive inhibition of rat CYP3A [38]. In contrast, its absence in DHT results in noncompetitive inhibition of rat CYP3A [38]. Furthermore, a pharmacophore model of TIwas developed and effectively used for screening potential inhibitors of spleen tyrosine kinase (SYK) from natural product databases containing 105,911 compounds through ligandpharmacophore screening [171].

## 7. Clinical Trials on Tanshinones Approved forHuman Treatment

Among the various tanshinones and their derivatives, only STS has received approval from the China Food and Drug Administration (CFDA) for the treatment of coronary heart disease and ischemic stroke [172,173,174,175]. Additionally, studies have demonstrated the safety and efficacy of STS as an adjunctive therapy for several conditions, including pulmonary heart disease, hypertensive nephropathy, ulcerative colitis, and acute viral myocarditis [176,177,178,179]. Furthermore, tanshinone-based compounds have been investigated for their potential therapeutic benefits in managing infantile hemangiomas of the skin and as an adjunctive treatment for angina pectoris [180,181].

## 8. Discussion

Analysis of the available data suggests that the mean C_max_ concentrations of tanshinones in the blood of humans or animals following oral administration range from 0.02 to 0.56 µM [21,22,27,50,52,54,55,57]. A higher mean C_max_ of 1.88 µM was observed after the intravenous administration of a water-soluble STS derivative [58]. The relatively low concentrations of tanshinones result in a limited number of protein targets that interact strongly enough to significantly block their biological functions [61,63,84,108,122,141]. This observation contrasts with available data suggesting that tanshinones are broadly involved in the regulation of numerous signaling pathways [32,45,150,151,156,158,161,162].

This contradiction may be partially explained by the extensive biological effects triggered by certain tanshinone targets, such as HuR and RNAPII [60,61,62,63]. Interaction with a single protein can lead to the altered expression of numerous genes, resulting in complex regulatory changes that ultimately influence protein concentrations or enzyme activities [60,61,63,155,182,183,184].

Other enzymes that could be directly and significantly inhibited by tanshinones at relatively low concentrations include TDO2, UGT1A9, MAGL, and 11β-HSD1 [84,108,122,141]. However, the limited number of these enzymes does not fully account for the widespread changes in the activity of numerous signaling pathways observed in humans and animals following oral tanshinone administration [32,45,151,152,155,157,158,159]. It can be hypothesized that different tanshinones may modulate the activity of various proteins within the same signaling pathway or simultaneously alter the functions of multiple pathways, leading to significant metabolic changes even at low ligand concentrations. This hypothesis is supported by the absence of a comprehensive list of proteins that could be targeted by tanshinones at concentrations achievable in vivo following oral administration. The incomplete nature of the available data is reflected in the broad and potent biological effects of tanshinones, as well as the numerous bioinformatic studies, which contrast with the relatively few high-affinity ligand–protein interactions that have been confirmed to date [40,41,42,43,44,45,168,185,186,187].

Another layer of complexity in tanshinone–protein interactions arises from the potential of tanshinones to reach much higher concentrations in organs such as the liver, intestine, or stomach compared to those observed in the blood [106]. As a result, the range of tanshinone targets and the biological processes affected in these organs may be broader than those presented in the review.

A promising starting point for future research could involve screening human or animal cDNA libraries to identify proteins that interact with tanshinones or their analogs. This approach would provide initial insights into the numerous protein targets active under in vivo conditions. Modified yeast three-hybrid (Y3H) systems, which are based on the yeast two-hybrid (Y2H) methodology, could serve as effective tools in these screenings [188,189,190,191]. Modified Y3H systems have been successfully employed to identify novel or confirm existing small molecule–ligand protein interactions under in vivo conditions [189,190,191]. Another potential approach involves the use of HPLC-based affinity chromatography to search for small molecule ligands in plant extracts, utilizing a selected protein or enzyme immobilized on glutaraldehyde-modified amino silica gel [192].

Affinity chromatography techniques, in combination with mass spectrometry, can be employed to systematically explore entire proteomes for proteins that interact with small molecule ligands [193,194,195,196,197]. The identified ligand–protein interactions must undergo rigorous validation and characterization through methods such as surface plasmon resonance, fluorescence resonance energy transfer, or enzyme inhibition kinetics and thermodynamics [98,108,109,198,199,200]. Detailed insights into tanshinone–protein interactions, obtained through crystal structure analysis or through precise, experimentally validated descriptions of ligand-binding sites, remain relatively scarce, highlighting a significant area for future research [201,202].

The low concentrations of tanshinone in human or animal blood following oral administration, coupled with a relative deficiency of well-characterized protein targets, can be attributed to the compound’s poor water solubility and limited bioavailability after oral intake [17,20,21]. Clearly established and currently utilized solutions to address these challenges include alternative administration routes, novel drug formulations, and the development of tanshinone analogs with improved pharmacokinetic profiles [22,23,24,25]. These strategies have led to substantial progress, resulting in a significant increase in tanshinone concentrations in the bloodstream [25,27,49,51,56]. As a result, the number of tanshinone–protein interactions detectable in animal models is increasing. Several tanshinone derivatives with enhanced water solubility and demonstrated inhibitory activity against human enzymes have been developed [29,30,31,203,204,205,206,207,208]. However, the safety and pharmacokinetic profiles of these derivatives in animals have not been extensively studied, limiting the comprehensive understanding of the biological effects they induce in living organisms. Therefore, significant progress is needed in the evaluation of the safety and pharmacokinetic properties of tanshinone analogs.

To date, only one component, tanshinone IIA sulfonate, has been approved in China for the treatment of cardiovascular diseases [172,173,174,175]. Available research suggests that tanshinones and their derivatives exhibit a wide range of biological activities, supporting their potential as drug leads for applications in cardiology, oncology, and neurology [32,83,84,116,151,154,159,160,163,176,177].

## Figures and Tables

**Figure 1 ijms-26-00848-f001:**
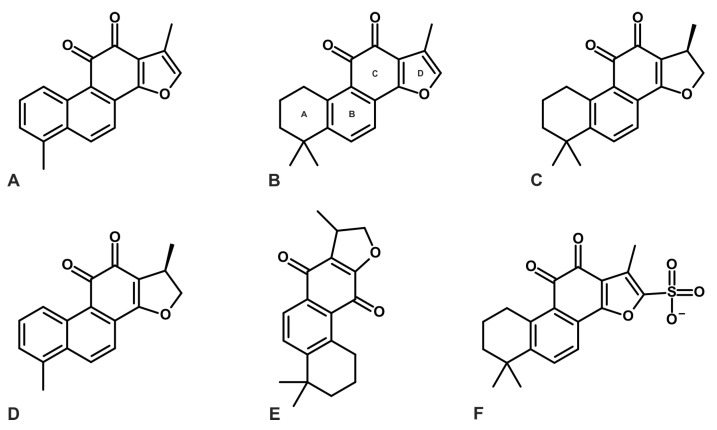
Chemical structures of key tanshinones and derivatives: (**A**) tanshinone I(TI), (**B**) tanshinone IIA (TIIA), (**C**) cryptotanshinone (CT), (**D**) dihydrotanshinone I (DHTI), (**E**) isocryptotanshinone, and (**F**) synthetic TIIA derivative tanshinone IIA sulphonate (STS), usually applied as a sodium salt (Na^+^ not presented on (**F**)). Tanshinone generally consists of four rings, including naphthalene or tetrahydronaphthalene rings A and B, a normal or paraquinone or lactone ring C, and a furan or dihydrofuran ring D [14,17].

**Figure 2 ijms-26-00848-f002:**
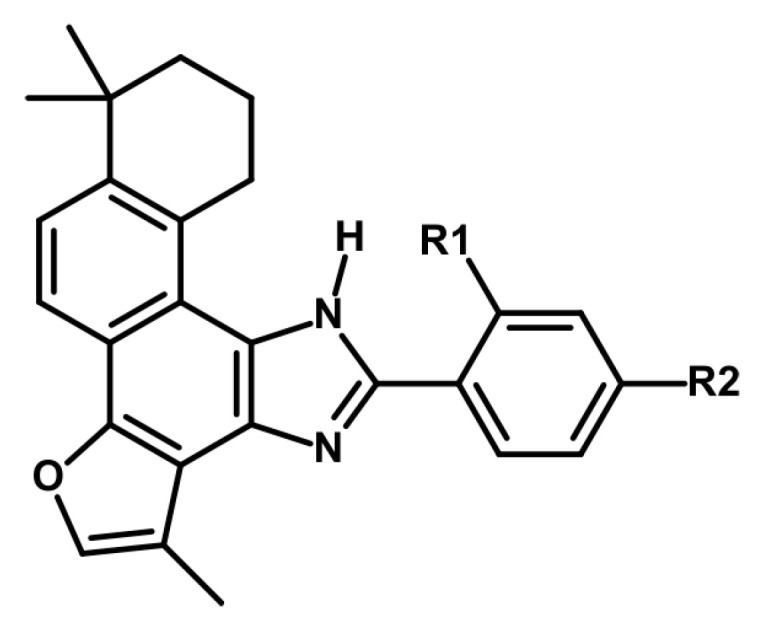
A schematic representation illustrating the inhibitory effects of eight TIIA derivatives on the growth of MDA-MB-231 breast cancer cells [65]. The derivatives synthesized include the following variations: Compound **1**: R_1_ = −CH_3_, R_2_ = −H; Compound **2**: R_1_ = −SO_2_CH_3_, R_2_ = −H; Compound **3**: R_1_ = −Cl, R_2_ = −H; Compound **4**: R_1_ = −Br, R_2_ = −H; Compound **5**: R_1_ = −H, R_2_ = −CH_3_; Compound **6**: R_1_ = −H, R_2_ = −SO_2_CH_3_; Compound **7**: R_1_ = −H, R_2_ = −Cl; Compound **8**: R_1_ = −H, R_2_ = −Br.

**Figure 3 ijms-26-00848-f003:**
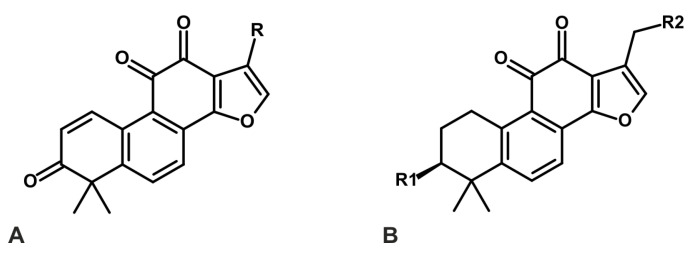
TIIA derivatives with strong inhibitory activity against IDO1 and TDO. (**A**) TIIA derivatives containing a 3-keto group, combined with different substituents (R = −CH_2_Cl, −CH_2_Br, −CH_2_OAc), show strong inhibitory activity against IDO1 and TDO. (**B**) Another group of TIIA derivatives, which strongly inactivate IDO1 and TDO, contain a 3-keto or 3-acetoxy group (R_1_ = =O or −OAc), with the C-17 methyl group substituted by hydrogen, hydroxyl, or acetoxyl groups (R_2_ = −OH, −H, −OAc) [83].

**Figure 4 ijms-26-00848-f004:**
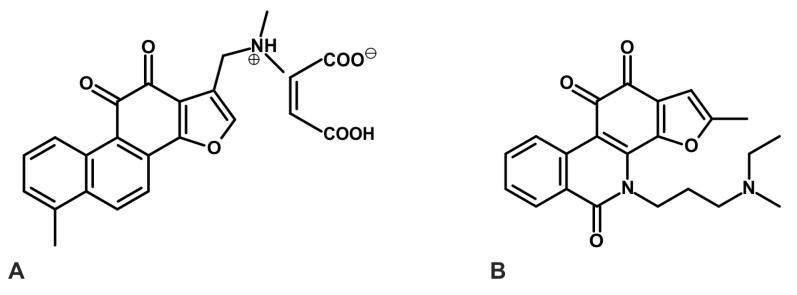
Two derivatives of TI, S439 (**A**) and S222 (**B**), directly inhibit the DNA topoisomerase I/II activity [86].

**Figure 5 ijms-26-00848-f005:**
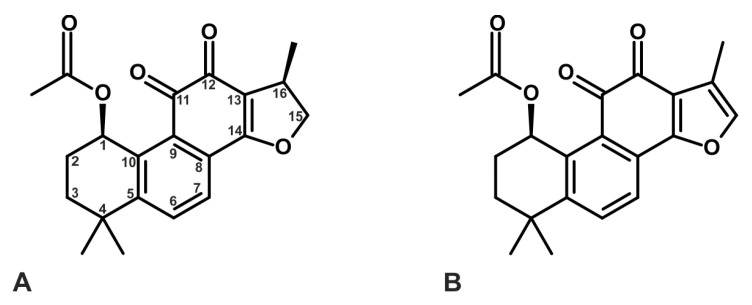
Structure of two naturally occurring tanshinones from *Perovskia atriplicifolia*: (**A**) (1*R*,15*R*)-1-acetoxycryptotanshinone and (**B**) (1*R*)-1-acetoxytanshinone IIA, which inactivate butyrylcholinesterase (BChE). Numbers 1–16 provide a position of carbon atoms in tanshinone skeleton [117].

**Table 1 ijms-26-00848-t001:** Mean C_max_ observed in selected pharmacokinetic studies on humans and animals.

Animal Studies			
Oral Administration of Pure Tanshinones, Plant Extracts or Plant Extract-Based Drugs			
Tanshinone	Range of Mean C_max_ in µg/L	Range of Mean C_max_ in µM	References
CT	20.89–48.8	0.07–0.16	[22,27,50,52,55]
DHTI	11.29–50.29	0.04–0.18	[22,50,52,55]
TI	10.1–77.01	0.03–0.26	[22,50,52,55]
TIIA	3.4–127.01	0.01–0.46	[22,50,51,52,54,55]
MT	40.79–144.90	0.14–0.51	[57]
Oral Administration of Solid Dispersions, Solid Lipid Nanoparticles or Micronized Tanshinoneformulations of Improved Bioavailability			
Tanshinone	Range of Mean C_max_ in µg/L	Range of Mean C_max_ in µM	References
CT	25.1–73.45	0.08–0.25	[27,49]
DHTI	None	None	-
TI	None	None	-
TIIA	75.26–165.36	0.26–0.56	[25,51,56]
Human Studies			
Traditional Decoction Containing 20 g of *S. miltiorrhiza*—Oral Route			
Tanshinone	Mean C_max_ in µg/L	Mean C_max_ in µM	References
CT	6.37	0.02	[21]
TI	0.43	0.002	[21]
TIIA	0.82	0.003	[21]
Micronized Preparation Containing 20 g of *S. miltiorrhiza*—Oral Route			
Tanshinone	Mean C_max_ in µg/L	Mean C_max_ in µM	References
CT	146.7	0.49	[21]
TI	6.57	0.02	[21]
TIIA	25.80	0.09	[21]
Synthetic Tanshinone Derivative—Intravenous Infusion			
Tanshinone	Mean C_max_ in µg/L	Mean C_max_ in µM	References
STS	743.6	1.88	[58]

This table summarizes the mean C_max_ values observed in pharmacokinetic studies for tanshinones and related compounds, calculated based on data presented in Appendix A. The conversion of C_max_ values from µg/L to µM utilized the molecular weights of the respective tanshinones: CT (296.40), DHTI (278.30), TIIA (294.30), I TI (276.30), STS (396.40), and MT (282.40).

**Table 2 ijms-26-00848-t002:** Proteins interacting with tanshinones and their derivatives, with IC_50_ values within or below those observed in animal and human studies described in Table 1.

Inhibited or Bound Enzyme	Mean IC_50_ Values of Enzyme Inhibition or Affinityto Enzyme	Reference
HuR	50 nM (affinity)	[61]
RNAPII	IC_50_ 0.2 µM Reduction of RNAPII occurrence within P2 promoter, gene intron and exon of *c-myc* oncogene	[63]
TDO2	IC_50_ 1 µM	[84]
UGT1A9	IC_50_ 0.27 µM	[108]
MAGL	IC_50_ 0.26 µM	[122]
11*β*-HSD1	Naturally observed tanshinone and tanshinone derivatives: IC_50_ for human 11*β*-HSD1 0.5–206.5 nM and mouse 11*β*-HSD1 0.4–392.3 nM	[141]

## Data Availability

No new data were created or analyzed in this study.

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
