# Peer review of "Proteins and DNA Sequences Interacting with Tanshinones and Tanshinone Derivatives"

_ijms, 2025, doi:10.3390/ijms26020848_

Round 1
Reviewer 1 Report
Comments and Suggestions for Authors
The manuscript entitled “Proteins and DNA Sequences Interacting with Tanshinones and Tanshinone Derivatives” summarizes the existing knowledge on interactions between tanshinones, their analogues, and biomolecules, including proteins and DNA. This review provides a foundation for deciphering the molecular mechanisms underlying the therapeutic effects of S. miltiorrhiza diterpenes. This paper generally meets the requirement of this journal. However, there are some issues in the paper that need to be addressed. Above all, I recommend this paper to be published after minor revision.
1. The clinically applied drugs of related tanshinones should be briefly summarized.
2. Since 40 tanshinones have been identified with diverse bioactivities, the structure-activity relationship of these tanshinones should be analyzed, especially for the key pharmacophores.
3. All the chemical structures in this paper should be unified in styles and ACS document 1996 is suugested.
4. Please carefully check the figures in Table 1, such as 0.03-0.26, 6.57?
5. Please check the expression of “IC50” through the whole manuscript. while “IC50 value” is correct. Such as Lines 290, 343, 345, 355, 358, 366, 372, 455-457, 470, 473, 511, 526, 530, 535, 539, 550, 554,
6. Please carefully check the figure of 1214 μM in Line 559.
7. Line 35, ranges from 0.269–1.137%---- ranges from 0.269% to 1.137%.
8. Lines 183 and 188, 6H---6H, 1H---1H
9. Line 226, tanshindiol B and C--- tanshindiols B and C
10. Lines 458 462, 463, 1R,15R---1R,15R.
11. Line 472, N-methyl---N-methyl
12. All the references should be carefully revised according to the styles of this journal.
Author Response
Response to comments provided by Reviewer 1
Authors are very grateful for all comments presented by Reviewer 1. The text was carefully corrected according to Reviewer 1 comments.
- The clinically applied drugs of related tanshinones should be briefly summarized.
Following chapter (numbered 7) was added to text – marked in red color.
- Clinical Trials on Tanshinones Approved to Human Treatment
Among the various tanshinones and their derivatives, only sodium tanshinone IIA sulfonate has received approval from the China Food and Drug Administration (CFDA) for the treatment of coronary heart disease and ischemic stroke [172,173,174,175]. Additionally, studies have demonstrated the safety and efficacy of sodium tanshinone IIA sulfonate as an adjunctive therapy for several conditions, including pulmonary heart disease, hypertensive nephropathy, ulcerative colitis, and acute viral myocarditis [176,177,178,179]. Furthermore, tanshinone-based compounds have been investigated for their potential therapeutic benefits in managing infantile hemangiomas of the skin and as an adjunctive treatment for angina pectoris [180,181].
- Since 40 tanshinones have been identified with diverse bioactivities, the structure-activity relationship of these tanshinones should be analyzed, especially for the key pharmacophores.
Following chapter (numbered 6) was added to text – marked in red color.
- Structure-activity Relationship of Tanshinones and their Derivatives
The anti-cancer effects of tanshinone I, tanshinone IIA, and six derivatives of tanshinone IIA on normal and cancerous colon cells have been analyzed, revealing that the naphthalene or tetrahydronaphthalene structures in rings A and B, along with the ortho-quinone moiety in ring C, are crucial for the bioactivity of tanshinones. Structural modifications in ring A and alterations to the furan or dihydrofuran groups in ring D were found to significantly influence activity [165]. Enhanced cytotoxicity was observed with increased delocalization within rings A and B, while a non-planar and compact D ring conferred improved anti-cancer activity. Structure-activity relationship (SAR) studies further indicated that the presence of polar and electron-withdrawing groups, such as -F, -NO2, -OH, or -CF3, at the para position of aromatic aldehydes significantly enhanced the activity. Conversely, substituting a -Br group in the furan ring of tanshinone IIA abolished its anti-cancer properties [165].
The findings align with those of Zhao et al. (2011), which demonstrated enhanced cytotoxicity of tanshinones following the introduction of polar substituents into the A or D rings [166]. The critical importance of an intact D ring, especially in its unsaturated form, was confirmed through cytotoxicity studies on H1299 and Bel-7402 cell lines. Furthermore, the presence of a 3-OH group has been identified as playing a significant role in the anti-cancer activity of tanshinones [167].
Structure-activity relationship (SAR) studies provide insights into the variations in antioxidant activity of tanshinones, which are mediated by the activation of nuclear factor erythroid 2–related factor 2 (Nrf2) [168]. A key mechanism for Nrf2 activation by quinones involves electrophilic modification of cysteine residues in Keap1, a critical negative regulator of Nrf2 (Abiko). Consequently, the higher electrophilic nature of tanshinones enhances their electron-abstracting capacity, leading to a greater potential to activate Nrf2 and stronger antioxidant properties [168,169].
Two parameters were employed to assess the electrophilic properties of tanshinones, namely electron affinity and the energy level of the lowest unoccupied molecular orbital (LUMO), in order to describe the indirect antioxidant activity of tanshinones and their derivatives [168]. Both parameters indicate that tanshinone I and dihydrotanshinone I exhibit stronger electron-acceptor properties compared to tanshinone IIA and cryptanshinone. The observed differences between these two groups of tanshinones are attributed to variations in the structure of ring A, with a benzene ring present in tanshinone I and dihydrotanshinone I, and a cyclohexane ring in tanshinone IIA and cryptanshinone [168]. Based on the Hammett constant, the electron-donating ability of methyl groups derived from cyclohexane is stronger than that of a benzene ring. Consequently, the more electrophilic properties of benzene-containing tanshinones may account for their enhanced potential to activate Nrf2. This mechanistic explanation also applies to the distinct structural differences between T-I, T-II-A, DHT, and CT. T-I and T-II-A feature a double bond in ring D, whereas DHT and CT possess only single bonds. According to the Hammett constant values, the methyl group is a stronger electron-donating group compared to the ethylene group. As a result, T-I and T-II-A exhibit greater activity than DHT and CT in electron abstraction, leading to subtle differences in Nrf2 activation and subsequent antioxidant properties [168].
Structural studies of tanshinones as agonists of the human estrogen receptor α ligand-binding domain (hERα-LBD) revealed that the binding affinity of tanshinones for hERα-LBD increases with their Connolly solvent-excluded volume (CSEV) [170]. Cryptotanshinone, having the largest volume, exhibits the strongest binding to the receptor, while tanshinone I, with the smallest volume, shows the weakest binding affinity for hERα-LBD. It is hypothesized that the larger compounds facilitate stronger hydrophobic interactions compared to smaller molecules [170].
Other structure-activity relationship (SAR) studies on tanshinones by Wang et al. (2011) demonstrated that the presence of a double bond at position 15 of the furan ring is associated with the competitive inhibition of rat CYP3A [38]. In contrast, its absence in dihydrotanshinone results in noncompetitive inhibition of rat CYP3A [38]. Furthermore, a pharmacophore model of Tanshinone I was developed and effectively used for screening potential inhibitors of spleen tyrosine kinase (SYK) from natural product databases containing 105,911 compounds through ligand-pharmacophore screening [171].
- All the chemical structures in this paper should be unified in styles and ACS document 1996 is suugested.
Figures are corrected as suggested.
- Please carefully check the figures in Table 1, such as 0.03-0.26, 6.57?
Corrected. The dot was included instead the comma.
- Please check the expression of “IC50” through the whole manuscript. while “IC50 value” is correct. Such as Lines 290, 343, 345, 355, 358, 366, 372, 455-457, 470, 473, 511, 526, 530, 535, 539, 550, 554,
Corrected. IC50 value or IC50 values were put in mentioned places.
- Please carefully check the figure of 1214 μM in Line 559.
The value is correct. Moreover, we added the corresponding value in mM (1.214 mM) in the bracket to assure this value..
- Line 35, ranges from 0.269–1.137%---- ranges from 0.269% to 1.137%.
Corrected.
- Lines 183 and 188, 6H---6H, 1H---1H
Corrected.
- Line 226, tanshindiol B and C--- tanshindiols B and C
Corrected.
- Lines 458 462, 463, 1R,15R---1R,15R.
Corrected.
- Line 472, N-methyl---N-methyl
Corrected.
- All the references should be carefully revised according to the styles of this journal.
Corrected. References were adjusted to the ijms journal requirements.
Reviewer 2 Report
Comments and Suggestions for Authors
This work by Szymczyk and co-workes summarizes the interactions of tanshinones with biomolecules, focusing on DNA and proteins.
It is a very fine work, very well written, and very well presented.
I have just a few of minor comments about the work I'm sure the authors can easily accomodate.
Lines 202 and following - while the reviews focuses on eukaryotic/human effects, zEZH2 and zSUZ12a/b appear a bit out of place. Soon after, the human forms of the enzymes are mentioned. Are there any equivalent studies on the human forms? Alternatively, what is similarity between the zebrafish and human forms?
Lines 323 and 445 to 475 - concentrations should come in molarity, according to the remainder of the text, for easier understanding by readers.
Line 570 - "in Table 1 and Table 2"
Table S2 is mentioned in line 579, but there's no use/discussion of its contents. Perhaps the authors could briefly discuss that these compounds effect on signalling pathways occurs through some gene regulation pathways (Neat, androgen receptor) but mostly through "classical" redox signalling and phosphorylation pathways.
One other aspect - a review should point forwards, building on the reviewed contents. The authors partially adress this regarding in the section 6 pointing out some studies that can be performed, but to what goal? what applications cand be foressen for these compounds/derivatives/future derivatives? I believe this would be highy hypothetical at the point, but could one envisage these compounds as drug leads for specific conditions?
Author Response
Response to comments provided by Reviewer 2
Authors are very grateful for all comments presented by Reviewer 2. The text was carefully corrected according to Reviewer 2 comments.
Lines 202 and following - while the reviews focuses on eukaryotic/human effects, zEZH2 and zSUZ12a/b appear a bit out of place. Soon after, the human forms of the enzymes are mentioned. Are there any equivalent studies on the human forms? Alternatively, what is similarity between the zebrafish and human forms?
Reply: Following fragment was added to the section 3 to more clearly characterize PRC2 complex, its components and evolutionary stability. Marked in red.
The PRC2 complex consists of enhancer of zeste homolog 2 (EZH2) or enhancer of zeste homolog 1 (EZH1), embryonic ectoderm development (EED), and suppressor of zeste 12 protein homolog (SUZ12) [67,68,69,70]. EZH2 functions as a histone methyltransferase, interacting with EED and SUZ12 to catalyze the transfer of a methyl group from the cofactor S-adenosylmethionine (SAM) to the ε-amino group of lysine residues [67,68,69,70]. Additionally, EED stabilizes the PRC2 complex, enhances its histone methyltransferase activity, and promotes chromatin expansion through trimethylation of H3K27, driven by EED–EZH2 interactions. EED further stabilizes the active site of PRC2 and provides a docking platform for factors that assemble the PRC2 complex, while SUZ12 interacts with EZH2 [67,68,69,70]. The components of the PRC2 complex are conserved across eukaryotes, suggesting the potential inhibitory activity of Tanshinone I (TI) against human homologs of EZH2 and SUZ12a/b [71,72,73].
Lines 323 and 445 to 475 - concentrations should come in molarity, according to the remainder of the text, for easier understanding by readers.
Reply: Corrected. Concentrations are calculated to the µM format.
Line 570 - "in Table 1 and Table 2"
Reply: Corrected.
Table S2 is mentioned in line 579, but there's no use/discussion of its contents. Perhaps the authors could briefly discuss that these compounds effect on signalling pathways occurs through some gene regulation pathways (Neat, androgen receptor) but mostly through "classical" redox signalling and phosphorylation pathways.
Reply. Following fragment was added to the section 5 to more precisely address issue presented in comment. Marked in red.
Among the affected signaling pathways are several, leading to the activation of NF-κB, androgen receptor, NEAT1, and cJUN, which modulate the expression of multiple genes [45,150,158,159,163,164].
One other aspect - a review should point forwards, building on the reviewed contents. The authors partially adress this regarding in the section 6 pointing out some studies that can be performed, but to what goal? what applications cand be foressen for these compounds/derivatives/future derivatives? I believe this would be highy hypothetical at the point, but could one envisage these compounds as drug leads for specific conditions?
Reply. Following fragment was addend at the end of Discussion section. Marked in red.
To date, the only one component, tanshinone IIA sulfonate, has been approved in China for the treatment of cardiovascular diseases [172,173,174,175]. Available research suggests that tanshinones and their derivatives exhibit a wide range of biological activities, supporting their potential as drug leads for applications in cardiology, oncology, and neurology. [32,83,84,116,151,154,159,160,163,176,177].